# Bateman gradients from first principles

Jussi Lehtonen [1✉]

In 1948, Angus Bateman presented experiments and concepts that remain influential and debated in sexual selection. The Bateman gradient relates reproductive success to mate number, and Bateman presented this as the cause of intra-masculine selection. A deeper causal level was subsequently asserted: that the ultimate cause of sex differences in Bateman gradients is the sex difference in gamete numbers, an argument that remains controversial and without mathematical backup. Here I develop models showing how asymmetry in gamete numbers alone can generate steeper Bateman gradients in males. This conclusion remains when the further asymmetry of internal fertilisation is added to the model and fertilisation is efficient. Strong gamete limitation can push Bateman gradients towards equality under external fertilisation and reverse them under internal fertilisation. Thus, this study provides a mathematical formalisation of Bateman's brief verbal claim, while demonstrating that the link between gamete number and Bateman gradients is not inevitable nor trivial.

[1] Department of Biological and Environmental Science, University of Jyväskylä, 40014 Jyväskylä, Finland. ✉email: jussi.lehtonen@iki.fi

In a classic but divisive 1948 article, Angus Bateman[1] presented experiments on *Drosophila melanogaster*, designed to address questions on intra-sexual selection and why it might differ between the two sexes. The lasting influence of the paper is due not only to empirical results but also to the conceptual framework Bateman introduced[2,3]. The most enduring and influential concept originating in the article is that of the 'Bateman gradient', the slope of reproductive success on the number of mates[3] (the concept was introduced by Bateman[1], named by Andersson and Iwasa[4], and given stronger theoretical underpinnings by many others (e.g., refs. [5–7].)). The Bateman gradient is a key component of the strength of sexual selection[3,5–7] and is typically expected to be steeper in males than females. Bateman described this difference in Bateman gradients between the sexes as the cause of stronger sexual selection in males than in females. Perhaps confusingly, a brief, purely verbal passage then claimed a deeper causal explanation: the ultimate cause of differences in intra-sexual selection is the difference in gamete number that exists between females and males (females being by definition those who make the larger and hence less numerous gametes: ref. [8]). In Bateman's words, "The primary cause of intra-masculine selection would thus seem to be that females produce much fewer gametes than males"[1]. For brevity, I shall call this 'Bateman's assertion'. The Bateman gradient concept is therefore intimately related to Bateman's assertion: the latter is a more fine-grained causal explanation for the former. However, Bateman's assertion remains a verbal claim with a dearth of explicit theory that directly links Bateman gradients to gamete numbers and anisogamy without invoking other intermediate sex-specific assumptions (a point of contention in sexual selection research in general[9]; see also e.g., ref. [10] for a discussion of criticisms of Bateman's article in particular). Models have been developed for Bateman gradients under various assumptions relating to reproductive biology[2,5,11], but without an explicit link to the gamete level. Conversely, recent models have linked anisogamy to stronger selection for sexually competitive traits in males than in females[12], and to the coevolution of parental investment and sexually selected traits in the two sexes[13], but included no explicit analysis of Bateman gradients. Here, I address this gap and show how a sex difference in Bateman gradients can arise from an imbalance in gamete numbers without invoking any other sex differences.

To be specific, I will not work directly with Bateman gradients, and instead, make use of 'Bateman functions'[3]. The Bateman function is a more general concept than the Bateman gradient, and for a given distribution of mating success, one can transition from a Bateman function to a Bateman gradient but not vice versa. The Bateman function has been defined as a mathematical function that represents the reproductive success of individuals with integer numbers of matings[3], and it therefore includes finer detail than the Bateman gradient which is just a single number. Here I will use a slightly amended definition for the Bateman function: a mathematical function that represents the average reproductive success of individuals with access to gametes from a given number of mates. A more gamete-centric definition engages directly with recent views on sexual selection[14,15], and with Bateman's assertion described above. A further reason for this amendment is that 'integer number of matings' suggests that Bateman functions and Bateman gradients are restricted to internal fertilisers, while they in fact can be applied equally well to broadcast spawners[16,17], plants[18], and other organisms without copulation, where the concept of 'mating' loses meaning and gamete access is not restricted to integer numbers of mates. For example, gametes from broadcast spawners can spread in partially overlapping clouds so that one individual can have access to gametes from e.g., 1.5 individuals of the opposite type. 'Number of mates' in the amended definition of the Bateman function can therefore refer to a continuous variable, not just integers.

Furthermore, when the Bateman function is specified for a continuous variable, it is consequently natural to consider the derivative of the Bateman function as an analogue of the Bateman gradient. The Bateman gradient and Bateman derivative are both measures of the slope of the mate number-reproductive success relationship (and under certain assumptions they have a simple relationship[5,18]). The Bateman derivative is a local measure of the slope of reproductive success on mate number $m$, while the Bateman gradient is a global statistical summary of the slope (Kokko, et al.[11] used a closely related formulation they termed the Bateman differential, defined as the derivative of fitness with respect to the mating rate). For the purposes of this article, it is sufficient to note that the Bateman derivative is analogous to a Bateman gradient, and both correspond visually to the steepness of the slope of the Bateman function; the Bateman functions derived below could be differentiated to find the corresponding Bateman derivatives, but for our purposes, a visual presentation provides a clearer and more intuitive link to Bateman gradients.

An additional central concept for the purposes of this article is that of a 'fertilisation function'. A fertilisation function is a mathematical function that models the number (or proportion) of successfully fertilised gametes based on gamete numbers or concentrations and additional parameters relating to the biology of the model organism[19]. Fertilisation functions have been developed for a wide range of biological scenarios and reproductive systems over several decades (e.g., refs. [20–24], reviewed in ref. [19]), but remain relatively rarely used in evolutionary theory. The definition of fertilisation functions suggests that they exist somewhere in the space between Bateman gradients and Bateman's assertion. Indeed, fertilisation functions will serve as a bridge between these concepts. The four central definitions are presented in Table 1.

In this work, I present analytical models where fertilisation functions are used to derive Bateman functions (and thus Bateman gradients) in a manner that invokes no sex-specific assumptions, or minimal ones where necessary. My initial goal is to construct models where the principal requirement is complete symmetry (Models 1–2). In other words, I seek hypothetical biological scenarios that are identical from the female and male perspectives, thus eliminating any potential sex-biased assumptions[9] in their biology. Furthermore, sex-biased assumptions are eliminated in the mathematical details by using methods that are also mathematically symmetrical, so that at any time it is possible to arbitrarily change the labels for males and females but reach the same conclusions. In other words, I derive a single equation that yields the Bateman function for both sexes simply by reversing the gamete number parameters. If such a function can replicate the salient features of Bateman gradients, Bateman's assertion of gamete numbers as the ultimate causal factor for Bateman gradient asymmetry has been formally validated. Such a model must, by definition, be based on external fertilisation, because internal fertilisation is an additional, asymmetrical assumption. I will then construct a model with the additional asymmetry of internal fertilisation (Model 3). The first two models confirm that gamete number asymmetry alone creates asymmetry in Bateman gradients as Bateman's assertion suggests, while the third model demonstrates that if fertilisation is efficient, this outcome remains valid despite the asymmetric roles of female as gamete recipient and male as gamete donor. However, inefficient fertilisation can push Bateman gradients towards equality under external fertilisation and reverse them under internal fertilisation. Overall, the results show that despite interesting exceptions, Bateman's assertion is correct under relatively general conditions.

**Table 1 Concepts and definitions.**

| | Pre- or post-ejaculatory: relating to the number of mates | | | Post-ejaculatory: relating to the number of gametes |
| --- | --- | --- | --- | --- |
| **Name** | **Bateman gradient** | **Bateman function** | **Bateman derivative** | **Fertilisation function** |
| Definition | Linear regression coefficient of reproductive success on the number of mates | Function that represents the average reproductive success of individuals with access to gametes from a given number of mates | Derivative of the Bateman function for the number of mates | Function that predicts the proportion or number of successfully fertilised gametes based on gamete numbers or concentrations |

## Results

### Model 1: Evolution of multiple mating and mate monopolisation under ancestral monogamy.

In all models, I assume a large population with a 1:1 sex ratio. I begin with what is possibly the simplest model set-up for deriving Bateman functions in a scenario that is completely symmetrical aside from gamete number. Assume a monogamous, externally fertilising population where parents pair up and release their gametes into a nest. That is, every individual in the initial population participates in exactly one fertilisation event (the equivalent of a mating). Now consider a mutant individual that can attract multiple mates of the opposite type to release gametes into its nest, with no competition from other individuals of its own type. This simple set-up avoids asymmetries arising from internal fertilisation, and the complication of direct gamete competition for the multiply mating mutant individual (which is examined in Models 2–3), placing focus directly on the core of the problem: the asymmetry arising in fertilisation from imbalanced gamete numbers. All gametes are released in one burst by all individuals, but the focal individual may achieve 'multiple matings' simply by monopolising multiple mates at its nest. The reproductive success of the focal individual is then equivalent to the number of fertilisations that take place in that nest. Our aim is to understand how the reproductive success of an individual deviating from the monogamous population strategy and instead mating with $\hat{m}$ individuals of the opposite type is altered. A strong positive relationship between $\hat{m}$ and reproductive success then indicates a steep Bateman gradient. If Bateman's assertion is correct, the resulting gradient should be steeper for the type that produces the larger number of gametes. Note that there is a game-theoretical[25] flavour to this setting, where the focus is on the fitness of a rare mutant in a population with a fixed resident strategy.

The two types are labelled with $x$ and $y$, which could correspond to the two sexes, depending on what gamete numbers are assigned to them. The number of gametes produced by a single individual is labelled $n_x$ and $n_y$, and the total number of gametes in a nest (or more generally, a fertilisation arena which could be internal or external) is labelled with $N_x$ and $N_y$. To compute the number of fertilisations in a nest with a total of $N_x$ and $N_y$ gametes, I use a fertilisation function first derived by Togashi et al.[24] purely from biophysical principles, treating the two gamete types symmetrically, with no pre-existing assumptions about differences between females and males or their gametes (for a broader context and comparison to other functions, see Table 1 and function $F_7$ in[19]). Any sex-specific differences arise only retrospectively after different gamete numbers are assigned to $x$ and $y$ of which either one could be male or female. The fertilisation function is $f(N_x, N_y) = N_x N_y \frac{e^{aN_x} - e^{aN_y}}{N_x e^{aN_x} - N_y e^{aN_y}}$, where $a$ is a parameter controlling fertilisation efficiency (for the special case $N_x = N_y$ the function is defined as $f(N_x, N_y) = \frac{aN_x^2}{1+aN_x}$[19,24], which is also the limit of $f$ when $N_y \rightarrow N_x$).

In a monogamous resident pair, we have simply $N_x = n_x$ and $N_y = n_y$. But if a mutant individual of type $x$ is able to attract $\hat{m}$ fertilisation partners of type $y$, then for that individual $N_y = \hat{m}n_y$, and the corresponding Bateman function is

$$b_x(\hat{m}) = f(N_x, N_y) = f(n_x, \hat{m}n_y) \quad (1)$$

where the fertilisation function $f$ is as described above. Because of symmetry, the corresponding function for $y$ is found simply by swapping $x$ and $y$. This function can reproduce the characteristic Bateman gradient asymmetry as gamete numbers diverge (progressing from isogamy to anisogamy in Fig. 1), showing how Bateman's assertion follows from biophysical effects that arise from unequal numbers of fusing particles: the fertilisation function $f$ is derived solely from such biophysical effects, not from any sex-specific assumptions. Equation (1) makes no reference to sexes, and they only become specified when values are assigned to $n_x$ and $n_y$. For example, if $n_x = 10$ and $n_y = 10,000$, the female Bateman function is $b_x(\hat{m})$ and the male Bateman function $b_y(\hat{m})$, where for the latter all $x$s in Eq. (1) are replaced with $y$s and vice versa. The labels $x$ and $y$ are truly just labels. While there are inevitably assumptions built into the equations, crucially we can be certain there are no sex-specific assumptions. Yet the typical shapes reminiscent of Bateman gradients arise from the model when different values are specified for $n_x$ and $n_y$ (Fig. 1).

Gamete limitation changes the results quantitatively so that under conditions of poor fertilisation efficiency a larger imbalance in gamete numbers is needed for Bateman gradients to diverge to a similar extent. However, even under inefficient fertilisation, the Bateman gradients do not reverse.

### Model 2: An external fertiliser model with population-level polygamy and gamete competition.

Model 1 presented the simplest possible scenario, where all individuals except a rare mutant mate only once, and gamete competition (sperm competition[26], but without assigning either gamete type to be sperm) was thus excluded for the focal mutant individual. Now I generalise from this to a situation that remains entirely symmetrical, but where the resident number of matings can take on any value, and then derive the Bateman function for a rare mutant that deviates from this population-level value. This set-up allows for gamete competition for the focal mutant individual, a crucial addition because of the empirical and theoretical importance of sperm competition[26], as well as earlier theory suggesting that polyandry decreases the sex difference in Bateman gradients[2].

The biological set-up is such that there is a large population and a large number of patches (fertilisation arenas) where multiple individuals of both sexes can release their gametes for fertilisation. After all individuals have released their gametes, those in each patch mix freely and fertilisations take place randomly. Set up in this way, the model is again identical from the perspective of both sexes, and gamete number can be isolated as the sole possible causal factor in any subsequent differences

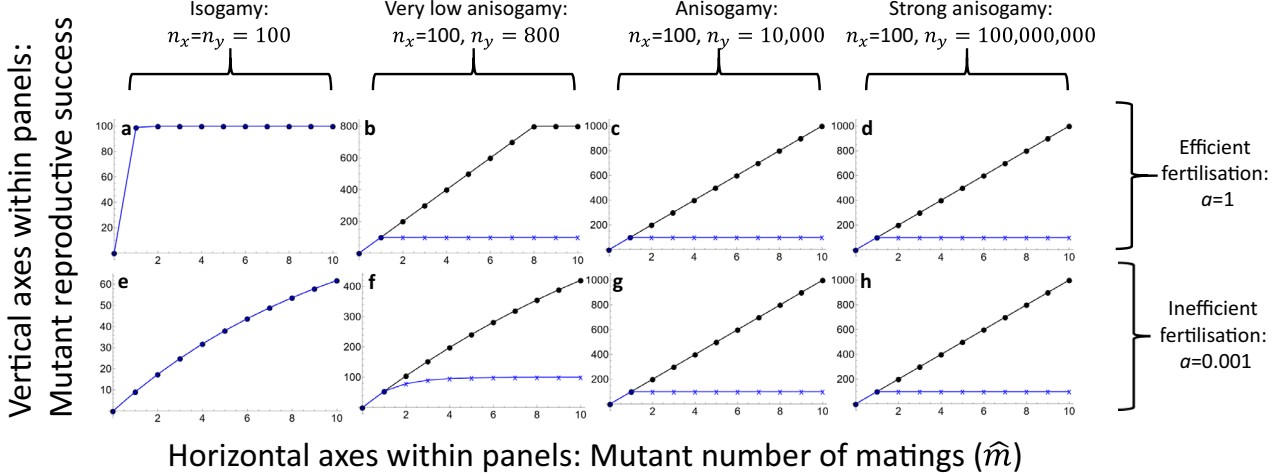

**Fig. 1 The Bateman function of Eq. (1).** This figure shows how the basic Bateman gradient asymmetry arises from simple biophysics and mathematics of fertilisation. The population is monogamous aside from a mutant individual, whose number of fertilisation partners ('matings') varies on the horizontal axes within panels. **a**–**d** show the effect of variation in sex-specific gamete numbers under efficient fertilisation, while **e**–**h** show the effect of variation in sex-specific gamete numbers under inefficient fertilisation. Parameter values used are shown in the figure. Females (gamete number $n_x$) are indicated by blue crosses and connecting lines, while males (gamete number $n_y$) are indicated by black dots and connecting lines. Under isogamy, females and males are undefined, and the two colours overlap. The typical sex-specific shapes of Bateman gradients arise from a single equation (which itself is not sex-specific) when a difference in gamete numbers is assigned to $n_x$ and $n_y$, confirming Bateman's intuition that the primary cause of the difference in selection is that females produce fewer gametes than males. The entire range of gamete number ratios presented in the figure is observed in nature, from equal gamete size in many unicellular organisms[39] to vertebrates, where sperm count per ejaculate can commonly exceed $10^9$ (see ref. [40] and Supplementary Information therein).

that may arise, extending from the initially monogamous and gamete competition-free scenario of Model 1. All individuals of both sexes are assumed to initially have the same strategy: to divide their $n_x$ or $n_y$ gametes equally between $m$ patches, and distribute themselves in such a way that gametes from $m$ individuals of each type release gametes into each patch (the number of individuals of each sex per patch need not necessarily be strictly equal to $m$, but this is the simplest assumption to account for the fact that gamete competition tends to increase with multiple 'matings'). Now, if a rare $x$ mutant divides its gametes evenly into $\hat{m}$ randomly selected patches, its gamete number per patch and consequently competitiveness in each patch is altered. Therefore, gametes of a mutant of type $x$ will gain, on average, a fraction $c_x = (n_x/\hat{m})/N_x$ of the fertilisations in that patch, where $N_x = n_x/\hat{m} + (m-1)n_x/m$. To compute the number of realised fertilisations in a patch, I use the same fertilisation function as in Model 1, where the mutant number of gametes in a patch is $N_x$ as above and the number of gametes of the opposite type is $N_y = m\frac{n_y}{m} = n_y$. All the components are now in place to write down the Bateman function corresponding to this scenario, for a mutant of type $x$:

$$b_x(\hat{m}, m) = \hat{m}c_x f\left(N_x, N_y\right) \quad (2)$$

where $c_x$, $N_x$ and $N_y$ are as defined above, and the fertilisation function $f$ is as in Model 1. For completeness, define $b_x(0, m) = 0$, which is necessarily true, but useful to define separately because division by 0 renders Eq. (2) formally undefined when $\hat{m} = 0$.

As in Model 1, Eq. (2) makes no reference to sexes, and they only become specified when values are assigned to $n_x$ and $n_y$ (Fig. 2).

**Model 3: An internal fertiliser model**. Models 1–2 were set up with the central aim of full symmetry and exclusion of any sex-specific assumptions. Internal fertilisation breaks this symmetry by introducing a sex-specific assumption other than gamete number.

Bateman gradients are, however, most commonly applied to situations with internal fertilisation where females are gamete recipients and males are gamete donors[27]. I therefore construct a model accounting for internal fertilisation. Where Eqs. (1) and (2) allowed no sex differences aside from gamete number, here I additionally consider the fact that females receive gametes while males donate them.

As in model 2, there is a very large population, and I assume that in the resident population, all females and males mate exactly $m$ times. It is then considered how a rare mutant individual's (of either sex) fitness depends on its number of matings $\hat{m}$.

I use the same fertilisation function as in Models 1-2. Consider first the female perspective (labelled with $x$). A female produces $n_x$ gametes and retains them internally. Each female mates with $m$ males, who also mate with $m$ females, dividing their gametes evenly over these matings. Therefore a mutant female receives $\hat{m}\frac{n_y}{m}$ male gametes, and her reproductive success is

$$b_x(\hat{m}, m) = f\left(n_x, \hat{m}\frac{n_y}{m}\right) \quad (3)$$

A mutant male, on the other hand, mates with $\hat{m}$ females, each of which mates with $m-1$ additional males. Therefore, the mutant male's mating partners will receive a total of $N_y = n_y/\hat{m} + (m-1)n_y/m$ male gametes. Thus, the mutant male gains a fraction $c_y = (n_y/\hat{m})/N_y$ of the fertilisations with each female, while the total reproductive success per female is $f(n_x, N_y)$. The mutant male's reproductive success is therefore

$$b_y(\hat{m}, m) = \hat{m}c_y f\left(n_x, N_y\right) \quad (4)$$

To avoid division by 0, we can again define $b_y(0, m) = 0$, analogous to Model 2. In contrast to Models 1–2, there are now separate equations for each sex because of the additional sex-specific assumption of internal fertilisation, but no further sex-specific assumptions are used in their derivation. Visually the Bateman functions (Fig. 3) are nevertheless very similar to Model

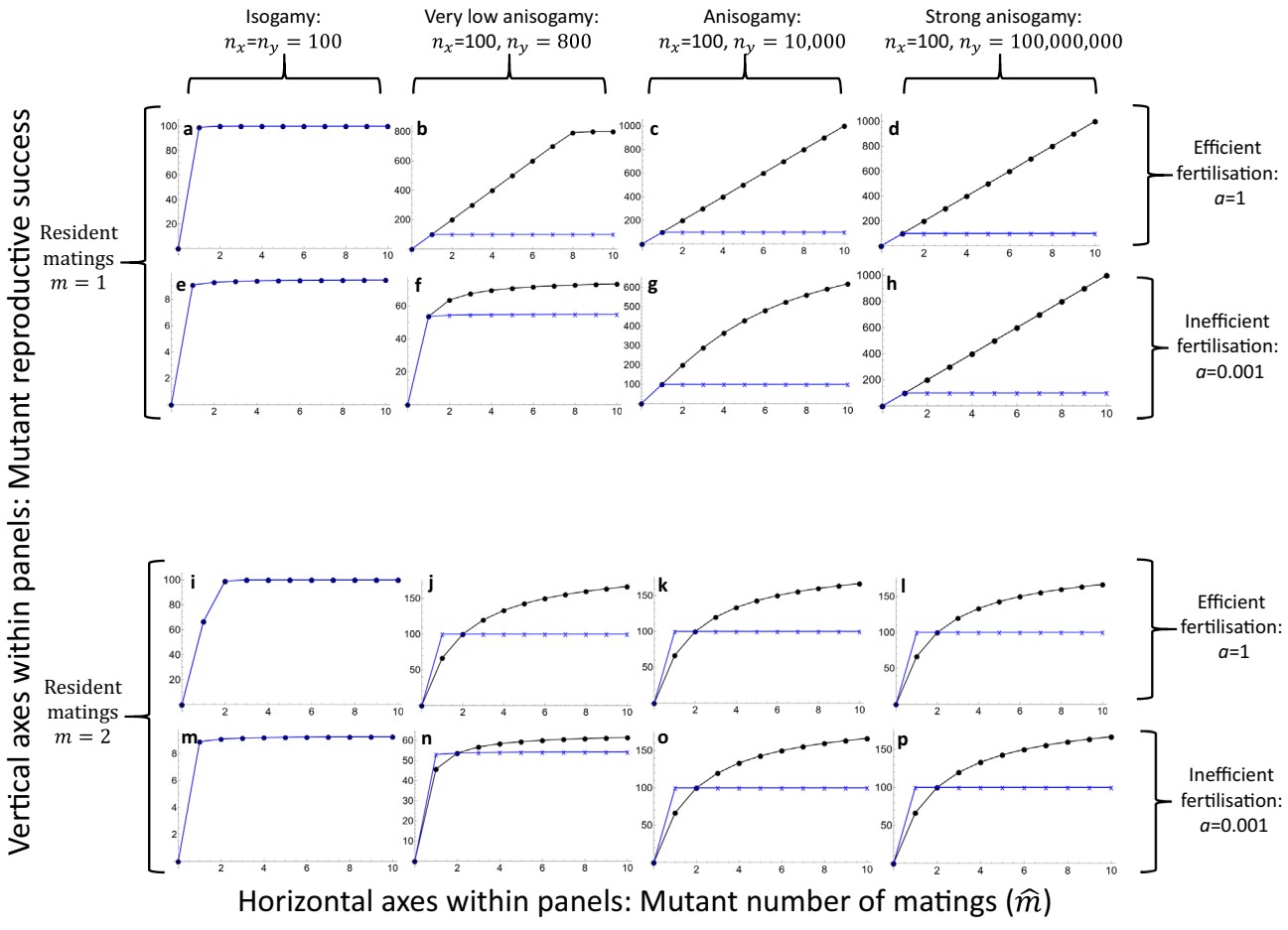

**Fig. 2 The Bateman function of Eq. (2) for an externally fertilising population with potential for population-wide polygamy and gamete competition.**
Results are shown for two values of resident matings ($m = 1$ and $m = 2$). **a**–**h** show the effect of variation in sex-specific gamete numbers and in fertilisation efficiency with $m = 1$, while **i**–**p** show the same with $m = 2$. Parameter values used are shown in the figure. The value $m = 2$ is used here because it is comparable to the mean number of matings in Bateman's[1] work (see Fig. 3 for corresponding results with internal fertilisation, but note that the aim of the models is not to quantitatively reproduce Bateman's results). Females (gamete number $n_x$) are indicated by blue crosses and connecting lines, while males (gamete number $n_y$) are indicated by black dots and connecting lines. Under isogamy, females and males are undefined, and the two colours overlap. Further variation in $m$ is examined in Fig. 4.

2, and again reproduce the sex-specific shapes first proposed by Bateman[1] when fertilisation is efficient. However, an interesting exception arises when relatively weak asymmetry in gamete numbers is combined with inefficient fertilisation and gamete limitation. When these conditions are combined with internal fertilisation, Bateman gradients can theoretically be reversed.

## Discussion

We have seen that completely symmetrical models of fertilisation and multiple mating can reproduce the salient features of Bateman gradients without invoking any sex-specific assumptions aside from the definitional gametic imbalance. The only sex-specific property included in Models 1 and 2 is a difference in gamete number, a fundamental property of the two sexes. Although the biological definition of the two sexes is commonly stated in terms of gamete size, a difference in gamete size translates to a difference in gamete number where the latter is often modelled as an inverse of gamete size, particularly in models of the ancestral origin of the two sexes[8] (it should be noted that in many contemporary organisms subsequent selection has led to a situation where total gamete volume is higher in females than in males[28,29], but gamete number is nevertheless higher in males). Here the central aim has been to analyse the

effect of a difference in gamete number while excluding all other sex differences and complications which are not directly relevant to the question. By excluding all other possible causes, I have validated Bateman's[1] assertion that a difference in gamete numbers between the sexes alone causes a difference in Bateman gradients between the sexes. In Model 3 I have included one sex-specific assumption beyond gamete number: internal fertilisation, such that females are gamete recipients and males are gamete donors, and the model shows that when fertilisation is efficient, conclusions drawn from the symmetrical models remain valid despite the introduction of this additional asymmetry.

Inefficient fertilisation (small parameter $a$ in the models) and consequently strong gamete limitation can alter the results under both external and internal fertilisation but in different ways. Gamete limitation can bring Bateman gradients back towards equality in external fertilisers even when gamete numbers are asymmetrical, but not reverse them (Figs. 1 and 2). A similar effect of diminishing the sex difference in Bateman gradients arises when the number of matings and polygamy increases at the population level (Fig. 4), in line with earlier theoretical results[2]. However, a combination of inefficient fertilisation and relatively small differences in gamete numbers between the sexes can theoretically reverse Bateman gradients under internal fertilisation (Fig. 3). Gamete limitation is not uncommon in external

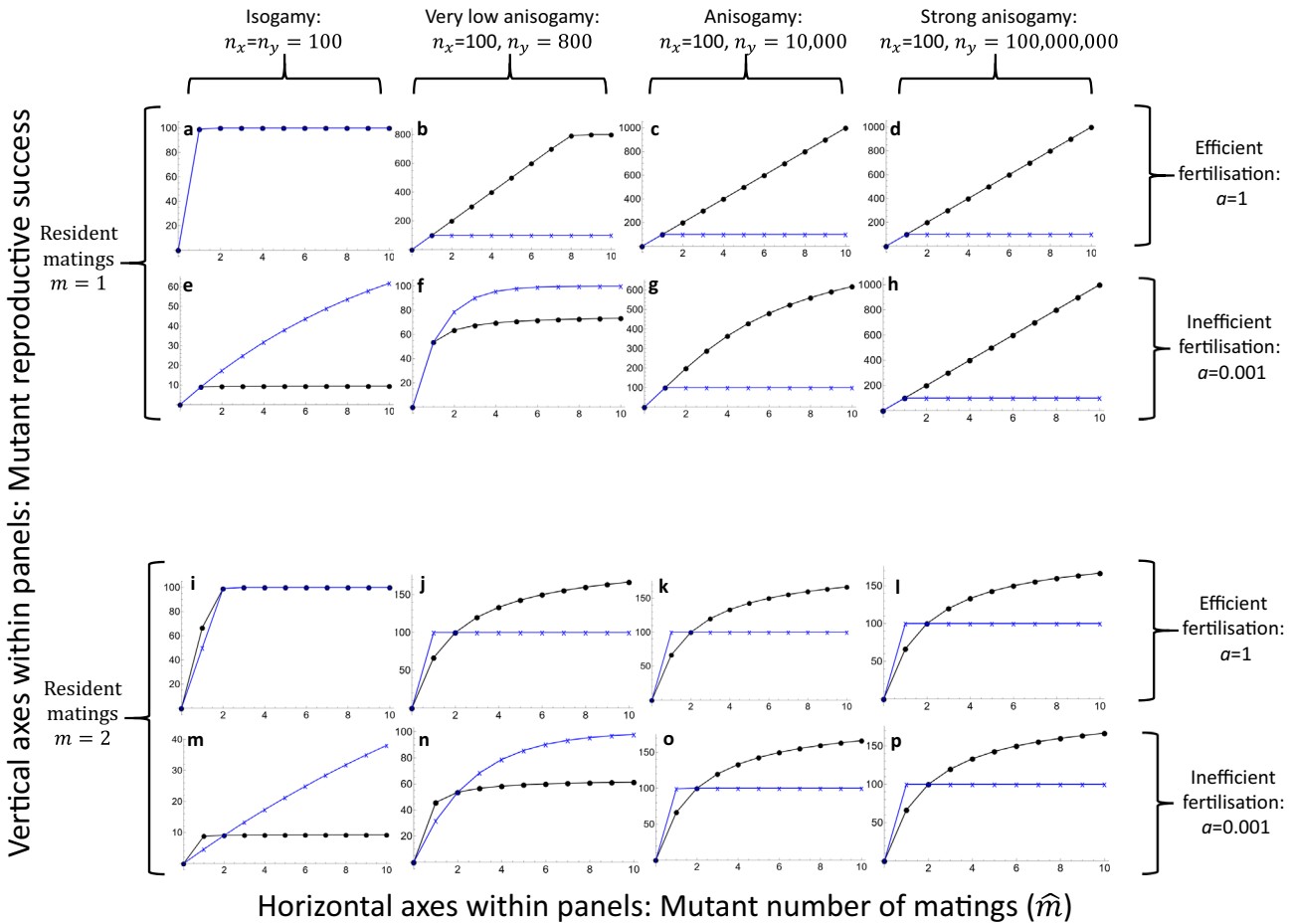

**Fig. 3 The Bateman functions of Eqs. (3) and (4) for internal fertilisation.** Where Figs. 1 and 2 show that the sex-specific shapes of Bateman functions are ultimately caused by differences in gamete number, Fig. 3 shows that internal fertilisation does not invalidate this outcome when fertilisation is efficient. As in Fig. 2, results are shown for two values of resident matings (1 and 2), and the value $m = 2$ is used because it is comparable to the mean number of matings in Bateman's[1] work. **a–h** show the effect of variation in sex-specific gamete numbers and in fertilisation efficiency with $m = 1$, while **i–p** show the same with $m = 2$. Parameter values used are shown in the figure. Inefficient fertilisation combined with relatively low asymmetry in gamete numbers and the added asymmetry of internal fertilisation can in principle reverse the Bateman gradients (second and fourth row). Females (gamete number $n_x$) are indicated by blue crosses and connecting lines, while males (gamete number $n_y$) are indicated by black dots and connecting lines.

fertilisers in nature[30,31], and the equalising effect seen in Figs. 1 and 2 may be significant for some external fertilisers. Similarly, the equalising effect seen in Fig. 4 seems plausible in polyandrous internally fertilising species. The reversal of Bateman gradients seen in Fig. 3 is more subtle. Sperm limitation is known to occur in internal fertilisers, but this is typically a consequence of male multiple mating[32]. The converse, where the evolution of multiple mating would be driven by sperm limitation in sperm recipients from a hypothetical ancestral state of monogamy then seems less likely and calls for further theoretical investigation. The novel observation that gamete limitation can in theory reverse Bateman gradients under internal fertilisation is therefore a very interesting one and shows that Bateman's seemingly simple assertion is far from trivial or obvious, even if broadly correct in its logic. Empirically, it has nevertheless been shown that Bateman gradients are steeper in males than in females in most animal species[27], although exceptions are not as uncommon as Bateman's writing might suggest[3,27] (in fairness, Bateman did claim the difference in Bateman gradients to be "almost universal"[1]).

A more abstract view of the models provides further insight into the source of the asymmetry in selection. First, note that all three models contain a fertilisation function $f$ and that typically a fertilisation function can be written using an alternative probabilistic notation: $f(N_x, N_y) = N_x p_x (N_x, N_y) = N_y p_y (N_x, N_y)$[19], or

written more concisely, $f = N_x p_x = N_y p_y$ where $p_x$ and $p_y$ indicate the per-gamete fertilisation probabilities of gametes of the two types. These equations imply that in any given fertilisation event the probability must be smaller for the more numerous gamete type and can indeed approach the maximum value of 1 for the less numerous gamete type[12]. It therefore seems intuitively plausible that in a situation that is otherwise symmetrical for the two sexes, the producer of the less numerous gametes (female) has less scope to increase this probability which explains the asymmetry in Figs. 1 and 2.

In the present models, there are two ways in which individuals can potentially increase $p_x$ or $p_y$ and thus their number of fertilised gametes. The focal individual can monopolise gametes from a larger number of opposite type individuals (Model 1 and the gamete recipient side of Model 3). Alternatively, the focal individual can spread its own gametes over a larger number of fertilisation events (Model 2 and the gamete donor side of Model 3). In both cases the absolute number of accessible gametes of the opposite type increases, and when fertilisation is efficient, the sex producing the larger number of gametes (males) has more to gain from this increase.

However, when fertilisation is inefficient, it is not just the absolute number of gametes that matters, but also their concentration or density. In Model 3, only females can increase the

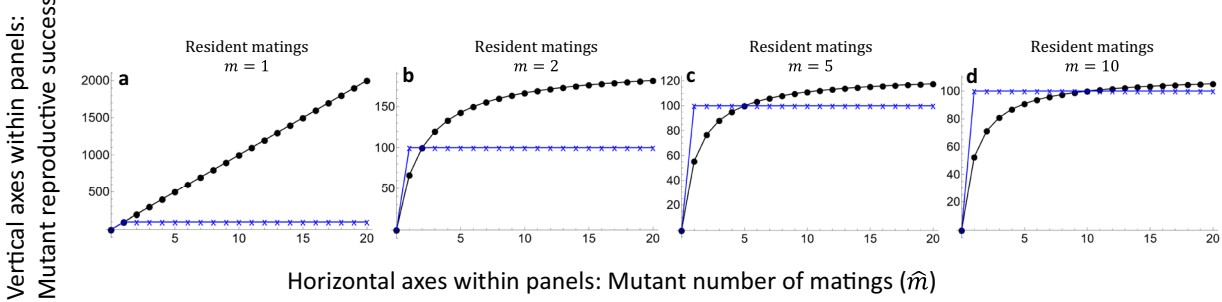

**Fig. 4 The Bateman functions of Eqs. (2)–(4) when the resident number of matings varies.** The gametic system is anisogamy with $n_x = 100$ (female, indicated by blue crosses and connecting lines), $n_y = 100,000,000$ (male, indicated by black dots and connecting lines). The number of resident matings $m$ varies between **a** - **d** as indicated above the panels. Results are visually indistinguishable for Models 2-3 and with fertilisation efficiency parameters $a = 0.001/a = 1$. Increased number of resident matings (i.e., increased gamete competition) decreases the steepness of the male Bateman gradient but does not eliminate the asymmetry between female and male gradients, in line with earlier theoretical results by Parker and Birkhead[2].

concentration of male gametes around their own gametes: when a female mates multiply in this model, the concentration of sperm around her eggs increases proportionally to the number of mates. When a male mates multiply in the same model, there is no such concentration effect, and instead, he dilutes his own gametes across a larger number of females whose egg concentration remains unchanged. This difference explains the reversal of Bateman gradients in Model 3 when fertilisation is inefficient: only females can improve fertilisation efficiency by mating multiply.

A somewhat ambiguous aspect of Bateman's[1] writing is the claim that the asymmetry arises from competition between male gametes for the fertilisation of the female gametes: "The primary cause of intra-masculine selection would thus seem to be that females produce much fewer gametes than males. Consequently, there is competition between male gametes for the fertilisation of the female gametes. And this competition is vastly more intense than that hitherto considered between zygotes". Bateman does not specify whether this means direct gamete competition or indirect competition for fertilisations. For example, Model 2 includes direct gamete competition among individuals of both types, whereas in Model 1 a mutant monopolising gamete of the opposite type faces no direct gamete competition, nor does a gamete recipient in Model 3. Selection, in general, requires competition in the sense of within-population variation in reproductive success, but from a logical perspective, the asymmetry in Bateman gradients does not necessitate direct competition between gametes of different individuals. In fact, polygamy and the resulting gamete competition tend to reduce (although not reverse) the difference between the male and female Bateman gradients (Fig. 4), in line with previous theoretical work[2]. A more general explanation for the difference in Bateman gradients is simply that the producer of the more numerous gametes has more to gain by increased access to opposite type gametes, irrespective of the presence of competing gametes from other individuals (but see above for exceptions when internal fertilisation, inefficient fertilisation, and relatively low ratios of gamete numbers are combined).

Bateman's work has been widely criticised in recent years[10,33–35]. Experimental methods have inevitably moved on over seven decades, making Bateman's approach outdated. Yet, as has been noted by others, Bateman's general conclusions are not necessarily negated by scrutiny of empirical methods, just as Mendel's experiments are not worthless despite their problems[2]. What is even more clear is that the conceptual framework initiated by Bateman's work retains value despite disagreements regarding experiments and their interpretation. This article has revisited one aspect of this conceptual framework, and one that Bateman based purely on verbal argument[1]: I have shown that Bateman's assertion relating gamete number to the Bateman

gradient and to sexual selection is correct under fairly general conditions, but not inevitable. Given that Bateman's assertion explicitly relates gamete numbers to reproductive success with no mathematical justification, a natural step forward is to use the mathematical machinery that has been developed for relating gamete numbers to fertilisation success since Bateman published his work—namely, fertilisation functions[19]. At the most general level, fertilisation functions can be derived from biophysical principles in a manner that is completely agnostic regarding sexes[19,24,36], allowing model construction that makes no sex-specific assumptions and thus avoids concerns relating to such assumptions and possible associated biases[9]. Any difference between the sexes arising in such a model must ultimately trace back to the gametic level. An additional purpose served by fertilisation functions here is that they permit modelling variation in fertilisation efficiency while maintaining consistency in the models (Figs. 1–3; note also that fertilisation efficiency could itself be causally linked to other factors, such as gamete size or motility[37], and the structure of the present models makes such potential future modifications straightforward).

Analysing the logical validity of Bateman's assertion is important for at least three reasons. Firstly, it shows that Bateman's assertion is far from obvious or trivial and that the argument is subtle, particularly under internal fertilisation. Second, it strengthens the mathematical foundations of Bateman's contested work, showing that despite exceptions, under fairly general conditions Bateman's assertion was correct. Third, it adds to our understanding of the 'sexual cascade' and the mainstream direction of selection in the evolutionary history of sexual reproduction[38], by showing why Bateman gradients are typically expected to diverge as a consequence of the evolution of anisogamy (which likely evolved under external fertilisation), thus linking Bateman gradients to the most fundamental biological definition of the two sexes[8]. Despite experimental methods that do not match the standards of this day, and despite verbal claims that were perhaps not as universal as the author suggested, Bateman's writing more than 70 years ago was remarkably prescient in explaining the causes of the mainstream flow of sexual differentiation.

## Methods
Models are described in the 'Results' section. The results presented in Figs. 1–4 arise directly from Eqs. (1)–(4).

## Data availability
No data was analysed or generated in this study. All results can be reproduced using the equations presented in the article.

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

## Acknowledgements

This research was funded by the Academy of Finland (grant number 340130, awarded to J.L.). J.L. would like to thank Geoff Parker for very helpful discussions and comments on the manuscript.

## Competing interests

The author declares no competing interests.
