## [Peer Review File · Nature Communications]

Bateman gradients from first principlesReviewers' Comments:

Reviewer #1:

Remarks to the Author:

This study, 'Bateman gradients from first principles', presents simple models linking the primordial sex difference (gamete size dimorphism = anisogamy) to sex differences in the Bateman gradient (which describes the relationship between the number of matings and reproductive success). This is a fundamental issue in the fields of sexual selection and mating system evolution, but it has become surprisingly controversial. I was overall quite pleased with this study. Through simple models, the authors establish a formal connection between anisogamy and sex differences in the Bateman gradient with as little baggage as possible. The models are simple enough that they could be used in textbooks and inform the debate more widely – a great strength in my view.

I found no issues with the mathematics. My main reservation was that the authors' explanations of their results sometimes seemed to ramble and also failed to intersect with some key literature. All of this seems entirely fixable. In particular, I would recommend revisiting the explanations on lines 348-372. In the internal fertilisation model, one fundamental difference between the male and female perspective is the way that gametes concentrate. When a female mates multiply, the concentration of sperm around her eggs increases proportionally to the number of mates. When a male mate's multiply, there is no such effect. I believe the authors tried to make this same point, but it was quite hard to follow their argument, especially in relation to the 'reversed' Bateman gradient. I also felt that the concept of 'local sperm competition' (*sensu* Schärer 2009, doi: 10.1111/j.1558-5646.2009.00669.x) might be helpful in understanding the results (see also minor comments below).

Some other literature that seemed important but missing:

- Theory on the relationship between anisogamy and sex roles, which implicitly relies on Bateman gradients even if it doesn't invoke the concept explicitly. E.g. Fromhage & Jennions (2016, doi:10.1038/ncomms12517)

- Theory on the Bateman gradient itself, e.g. Jones (2009, doi: 10.1111/j.1558-5646.2009.00664.x) Henshaw et al. (2018, doi: 10.1111/evo.13554)

- Literature on the relationship between gamete size and fertilisation efficiency (the authors do not consider this relationship, which is fine, but there is an established literature on it and some of the authors' conclusions might change under different assumptions). E.g. Dusenbery (2011, doi:10.1017/CBO9780511975943.007)

Minor comments:

L46-47: 'one can always transition from a Bateman function to a Bateman gradient but not vice versa': This makes it sound as though a Bateman function contains all of the information necessary to construct the Bateman gradient, but this is not true, as the Bateman gradient also depends on the distribution of mating success. A single Bateman function can correspond to more than one Bateman gradient (i.e. a many-to-many rather than a one-to-many relationship).

L58-61: 'once we dispense the requirement for an integer number of matings, the function is specified for a continuous variable, and it is consequently natural to consider the derivative of the Bateman function as an analogue of the Bateman gradient': I can imagine constructions along these lines, but above you define the Bateman function as 'the average reproductive success of individuals with access to gametes from a given number of mates'. Surely the latter is still an integer?

L145: 'the function becomes': I would say 'the function is defined as', as strictly speaking the previous

function is indeterminate when $N_x = N_y$.

Fig. 1 labels: I have no issues with the choice of parameter values, but labelling the third column as 'low anisogamy' is not quite in line with the empirical situation. Modest anisogamy ratios (e.g., 1000 to 1, like here) are empirically fairly common, especially when eggs are broadcast in great numbers. I would label the third column 'anisogamy' and the fourth 'advanced anisogamy' or something.

L210: Typo 'gametes m individuals'. Also I suggest 'both types' -> 'each type'. In the next line, specify 'of each sex' or change to $2m$.

Eq. 4: It's a matter of taste, but I would cancel the m -hats on the right-hand side.

L276: 'Contrary' -> 'In contrast'

L309-310: 'a difference in gamete size translates to a difference in gamete number, where the latter is often modelled as an inverse of gamete size': It's often modelled that way, but empirically it's rarely that simple. Total gamete volume is typically higher in females than in males, for instance. See e.g. Hayward & Gillooly (2011, doi:10.1371/journal.pone.0016557) Parker et al. (2018, doi:10.1111/brv.12363) Some additional explanation / caution would be helpful here.

L369-370: 'a multiply mating donor tends to only makes the situation worse for recipients by diluting its gametes over multiple matings': I don't see how this is relevant if we are talking about the fitness interests of the donor.

L370-372: 'At the same time, the donor is unable to significantly increase its own fitness by mating multiply, because of the gamete limited state of each of its mating partners': While true, this doesn't really help the reader. I think the issue is that when fertilisation is inefficient, there is effectively no competition among the sperm of a single individual (no 'local sperm competition') or indeed among sperm of multiple individuals. So spreading the sperm out over multiple partners doesn't change anything.

Reviewer #2:

Remarks to the Author:

This paper aims to confirm Bateson intuition that the ultimate cause of sex differences, in what sexual selection is concerned, arises from sex differences regarding gametes number. Using three different models, the authors conclude that Bateman's assertion is correct.

I cannot add much that would improve this work. The paper is well-written, the mathematical models appropriate for the question, and conclusions arise naturally from the results of the models. They need, however, to indicate the difference between the black lines and grey lines, as it took me a while to understand the figures.

Regardless, I couldn't recommend more this paper to be published in Nature Communications. I have repeatedly heard people in conferences saying how Bateson's conclusions are wrong - usually using simple verbal statements. The current paper vindicates Bateson's argument through mathematical models, a more robust tool to evaluate theoretical ideas.

REVIEWER COMMENTS

Reviewer #1 (Remarks to the Author):

This study, 'Bateman gradients from first principles', presents simple models linking the primordial sex difference (gamete size dimorphism = anisogamy) to sex differences in the Bateman gradient (which describes the relationship between the number of matings and reproductive success). This is a fundamental issue in the fields of sexual selection and mating system evolution, but it has become surprisingly controversial. I was overall quite pleased with this study. Through simple models, the authors establish a formal connection between anisogamy and sex differences in the Bateman gradient with as little baggage as possible. The models are simple enough that they could be used in textbooks and inform the debate more widely – a great strength in my view.

Thank you for these encouraging words.

I found no issues with the mathematics. My main reservation was that the authors' explanations of their results sometimes seemed to ramble and also failed to intersect with some key literature. All of this seems entirely fixable. In particular, I would recommend revisiting the explanations on lines 348-372. In the internal fertilisation model, one fundamental difference between the male and female perspective is the way that gametes concentrate. When a female mates multiply, the concentration of sperm around her eggs increases proportionally to the number of mates. When a male mate's multiply, there is no such effect. I believe the authors tried to make this same point, but it was quite hard to follow their argument, especially in relation to the 'reversed' Bateman gradient. I also felt that the concept of 'local sperm competition' (sensu Schärer 2009, doi: 10.1111/j.1558-5646.2009.00669.x) might be helpful in understanding the results (see also minor comments below).

The original text could indeed have been clearer, and the reviewers suggestion of framing the explanation via gamete concentration is excellent, concise, and clear. The indicated lines have been rewritten. In particular, the explanation using gamete concentration now reads:

"However, when fertilisation is inefficient, it is not just the absolute number of gametes that matters, but also their concentration or density. In Model 3 only females can increase the concentration of male gametes around their own gametes: when a female mates multiply in this model, the concentration of sperm around her eggs increases proportionally to the number of mates. When a male mates multiply in the same model, there is no such concentration effect, and instead he dilutes his own gametes across a larger number females whose egg concentration remains unchanged. This difference explains the reversal of Bateman gradients in Model 3 when fertilisation is inefficient: only females can improve fertilisation efficiency by mating multiply."

Regarding local sperm competition, see the response to the last minor comment below.

Some other literature that seemed important but missing:

- Theory on the relationship between anisogamy and sex roles, which implicitly relies on Bateman gradients even if it doesn't invoke the concept explicitly. E.g. Fromhage & Jennions (2016, doi:10.1038/ncomms12517)

Now cited as reference 13.

- Theory on the Bateman gradient itself, e.g. Jones (2009, doi: 10.1111/j.1558-5646.2009.00664.x) Henshaw et al. (2018, doi: 10.1111/evo.13554)

Now cited as references 6 and 7.

- Literature on the relationship between gamete size and fertilisation efficiency (the authors do not consider this relationship, which is fine, but there is an established literature on it and some of the authors' conclusions might change under different assumptions). E.g.

Dusenbery (2011, doi:10.1017/CBO9780511975943.007)

Now cited as reference 37, with the following explanation: "Figs 1-3; note also that fertilisation efficiency could itself be causally linked to other factors, such as gamete size or motility³⁷, and the structure of the present models makes such potential future modifications straightforward"

Minor comments:

L46-47: 'one can always transition from a Bateman function to a Bateman gradient but not vice versa': This makes it sound as though a Bateman function contains all of the information necessary to construct the Bateman gradient, but this is not true, as the Bateman gradient also depends on the distribution of mating success. A single Bateman function can correspond to more than one Bateman gradient (i.e. a many-to-many rather than a one-to-many relationship).

This is a very good point. The sentence now reads ". The Bateman function is a more general concept than the Bateman gradient, and for a given distribution of mating success, one can transition from a Bateman function to a Bateman gradient but not vice versa."

L58-61: 'once we dispense the requirement for an integer number of matings, the function is specified for a continuous variable, and it is consequently natural to consider the derivative of the Bateman function as an analogue of the Bateman gradient': I can imagine constructions along these lines, but above you define the Bateman function as 'the average reproductive success of individuals with access to gametes from a given number of mates'. Surely the latter is still an integer?

An explanation has been added for why this need not be an integer: "A further reason for this amendment is that 'integer number of matings' suggests that Bateman functions and Bateman gradients are restricted to internal fertilisers, while they in fact can be applied equally well to broadcast spawners^{16,17}, plants¹⁸, and other organisms without copulation, where the concept of 'mating' loses meaning and gamete access is not restricted to integer numbers of mates. For example, gametes from broadcast spawners can spread in partially overlapping clouds so that one individual can have access to gametes from e.g., 1.5 individuals of the opposite type. 'Number of mates' in the amended definition of the Bateman function can therefore refer to a continuous variable, not just integers."

L145: 'the function becomes': I would say 'the function is defined as', as strictly speaking the previous function is indeterminate when $N_x = N_y$.

Rewritten exactly as suggested.

Fig. 1 labels: I have no issues with the choice of parameter values, but labelling the third column as 'low anisogamy' it not quite in line with the empirical situation. Modest anisogamy ratios (e.g., 1000 to 1, like here) are empirically fairly common, especially when eggs are broadcast in great numbers. I would label the third column 'anisogamy' and the fourth 'advanced anisogamy' or something.

A good point. The labels have been changed to "Isogamy", "Very low anisogamy", "Anisogamy", "Strong anisogamy".

L210: Typo 'gametes m individuals'. Also I suggest 'both types' -> 'each type'. In the next line, specify 'of each sex' or change to 2m.

Changed as suggested, and the full revised text reads: "All individuals of both sexes are assumed to initially have the same strategy: to divide their n_x or n_y gametes equally between m patches, and distribute themselves in such a way that gametes from m individuals of each type release gametes into each patch (the number of individuals of

each sex per patch need not necessarily be strictly equal to m , but this is the simplest assumption to account for the fact that gamete competition tends to increase with multiple 'matings')."

Eq. 4: It's a matter of taste, but I would cancel the m -hats on the right-hand side.

This is indeed a trade-off and matter of taste: cancelling the m -hats would be cleaner mathematically, while not cancelling retains the causal structure of the equation more intact and readable. As a compromise, the last part of the equation has been removed, so that the components are not spelled out in full in the equation itself. The equation remains fully readable, because all components are written in the preceding paragraph, and the shorter structure comes with two benefits: it is in line with all other equations in the article, which are written in this abbreviated way; and there are now no m -hats to cancel, while the causal structure of the equation remains clear. The equation now reads " $b_y(\hat{m}, m) = \hat{m}c_y f(n_x, N_y)$ (4)".

L276: 'Contrary' -> 'In contrast'

Fixed.

L309-310: 'a difference in gamete size translates to a difference in gamete number, where the latter is often modelled as an inverse of gamete size': It's often modelled that way, but empirically it's rarely that simple. Total gamete volume is typically higher in females than in males, for instance. See e.g. Hayward & Gillooly (2011, doi:10.1371/journal.pone.0016557) Parker et al. (2018, doi:10.1111/brv.12363) Some additional explanation / caution would be helpful here.

Agreed, and some explanation / caution has now been added (28 and 29 are the suggested references): "The only sex-specific property included in Models 1 and 2 is a difference in gamete number, a fundamental property of the two sexes. Although the biological definition of the two sexes is commonly stated in terms of gamete size, a difference in gamete size translates to a difference in gamete number where the latter is often modelled as an inverse of gamete size, particularly in models of the ancestral origin of the two sexes ⁸ (it should be noted that in many contemporary organisms subsequent selection has led to a situation where total gamete volume is higher in females than in males ^{28,29}, but gamete number is nevertheless higher in males). Here the central aim has been to analyse the effect of a difference in gamete number, while excluding all other sex differences and complications which are not directly relevant to the question. By excluding all other possible causes, we have validated Bateman's ¹ assertion that a difference in gamete numbers between the sexes alone causes a difference in Bateman gradients between the sexes."

L369-370: 'a multiply mating donor tends to only makes the situation worse for recipients by diluting its gametes over multiple matings': I don't see how this is relevant if we are talking about the fitness interests of the donor.

This section has been rewritten (see response to the reviewers first suggestion above).

L370-372: 'At the same time, the donor is unable to significantly increase its own fitness by mating multiply, because of the gamete limited state of each of its mating partners': While true, this doesn't really help the reader. I think the issue is that when fertilisation is inefficient, there is effectively no competition among the sperm of a single individual (no 'local sperm competition') or indeed among sperm of multiple individuals. So spreading the sperm out over multiple partners doesn't change anything.

This section has been rewritten in response to the first suggestion by this reviewer (above).

Local sperm competition (or local gamete competition) is a useful concept and term, but it does not seem ideal as an explanation here. The relevant original paragraphs were already too long and difficult to follow (as noted by reviewer), and they have now been shortened using the reviewer's very helpful suggestion of focusing on gamete concentration.

*Using local gamete competition as an explanation would require 1) explaining what local gamete competition is, 2) explaining why gamete limitation leads to diminished local gamete competition and 3) explaining why diminished local gamete competition diminishes benefits from spreading sperm out over multiple partners. The logic is quite subtle. For example, in the model of Henshaw et al 2014 (Local Gamete Competition Explains Sex Allocation and Fertilization Strategies in the Sea, AmNat) gamete limitation results in **increased** local gamete competition among eggs. The scenario, models, and focus are different, but while it may be obvious to some why in the present model local gamete competition would be diminished, it probably would not be obvious to most readers.*

The suggestion was taken seriously, but attempts to write local sperm competition into the explanation resulted in text that rambled even more than the original (see first suggestion by this reviewer and the response above), and as a compromise the paragraph has been shortened and clarified using the suggested focus on gamete concentration, but without using the local sperm competition concept.

Reviewer #2 (Remarks to the Author):

This paper aims to confirm Bateson intuition that the ultimate cause of sex differences, in what sexual selection is concerned, arises from sex differences regarding gametes number. Using three different models, the authors conclude that Bateman's assertion is correct.

I cannot add much that would improve this work. The paper is well-written, the mathematical models appropriate for the question, and conclusions arise naturally from the results of the models. They need, however, to indicate the difference between the black lines and grey lines, as it took me a while to understand the figures.

Regardless, I couldn't recommend more this paper to be published in Nature Communications. I have repeatedly heard people in conferences saying how Bateson's conclusions are wrong - usually using simple verbal statements. The current paper vindicates Bateson's argument through mathematical models, a more robust tool to evaluate theoretical ideas.

Many thanks for these positive and encouraging comments. As suggested, the difference between black and grey lines has now been indicated in the legend to Fig. 1.

Reviewers' Comments:

Reviewer #1:

Remarks to the Author:

I thank the authors for their revisions, which I feel have resolved all reservations I originally had about the manuscript. I think this is a very nice contribution to the debate about the implications of anisogamy for the evolution of sex roles.

I have one small note - the legends to Fig. 1 and 2 were changed in the lower halves of these figures, but not the upper halves.

REVIEWERS' COMMENTS

Reviewer #1 (Remarks to the Author):

I thank the authors for their revisions, which I feel have resolved all reservations I originally had about the manuscript. I think this is a very nice contribution to the debate about the implications of anisogamy for the evolution of sex roles.

I have one small note - the legends to Fig. 1 and 2 were changed in the lower halves of these figures, but not the upper halves.

I am very happy to hear that the revisions have resolved all reservations the reviewer had with the previous version. I thank the reviewer for their suggestions which undoubtedly improved the manuscript.

Regarding the figure legends: I believe the reviewer's comment refers to the 'track changes' version of the resubmitted manuscript, where the new versions of the figures appear below the old ones. In the submitted final figure files all legends have been changed.